# Good Artists Copy, Great Artists Steal: Model Extraction Attacks Against Image Translation Models

## Abstract

Machine learning models are typically made available to potential client users via inference APIs. *Model extraction attacks* occur when a malicious client uses information gleaned from queries to the inference API of a victim model $F_{\mathcal{V}}$ to build a *surrogate model $F_{\mathcal{A}}$* with comparable functionality. Recent research has shown successful model extraction of image classification, and natural language processing models.

In this paper, we show the first model extraction attack against real-world generative adversarial network (GAN) *image translation models*. We present a framework for conducting such attacks, and show that an adversary can successfully extract functional surrogate models by querying $F_{\mathcal{V}}$ using data from the same domain as the training data for $F_{\mathcal{V}}$. The adversary need not know $F_{\mathcal{V}}$'s architecture or any other information about it beyond its intended task.

We evaluate the effectiveness of our attacks using three different instances of two popular categories of image translation: (1) Selfie-to-Anime and (2) Monet-to-Photo (image style transfer), and (3) Super-Resolution (super resolution). Using standard performance metrics for GANs, we show that our attacks are effective. Furthermore, we conducted a large scale (125 participants) user study on Selfie-to-Anime and Monet-to-Photo to show that human perception of the images produced by $F_{\mathcal{V}}$ and $F_{\mathcal{A}}$ can be considered equivalent, within an equivalence bound of Cohen's $d = 0.3$.

Finally, we show that existing defenses against model extraction attacks (watermarking, adversarial examples, poisoning) do not extend to image translation models.

## 1 Introduction

Machine learning (ML) models have become increasingly popular across a broad variety of application domains. They range from tasks like image classification and language understanding to those with strict safety requirements like autonomous driving or medical diagnosis. Machine learning is a multi-billion dollar industry (TechWorld, 2018) supported by technology giants, such as Microsoft, Google, and Facebook.

Recently *image translation* applications have become popular in social media. Examples include coloring old photos (Isola et al., 2017), applying cartoon based filters (Rico Beti) or generating fake images of people (Deep-Fakes (Tolosana et al., 2020)). Features like face filters or face transformations are now an integral part of various popular applications such as TikTok, Snapchat or FaceApp. Such features are costly to create. They require data collection and sanitization, engineering expertise as well as computation resources to train *generative adversarial network* (GAN) models to implement these features. This represents a high barrier-to-entry for newcomers who want to offer similar features. Hence, effective image translation models confer a business advantage on their owners.

Use of ML models is typically done in a *black-box* fashion – clients access the models via inference APIs, or the models are encapsulated within on-device sandboxed apps. However, a malicious client (adversary) can mount *model extraction attacks* (Tramèr et al., 2016) by querying a model and using the responses to train a local *surrogate model* $F_{\mathcal{A}}$ that duplicates the functionality of the victim model $F_{\mathcal{V}}$. So far model extraction attacks have been shown to be successful in stealing image classifiers (Tramèr et al., 2016; Papernot et al., 2017; Juuti et al., 2019; Orekondy et al., 2019; Correia-Silva et al., 2018; Jagielski et al., 2020), and natural language processing models (Krishna et al., 2020; Wallace et al., 2021).

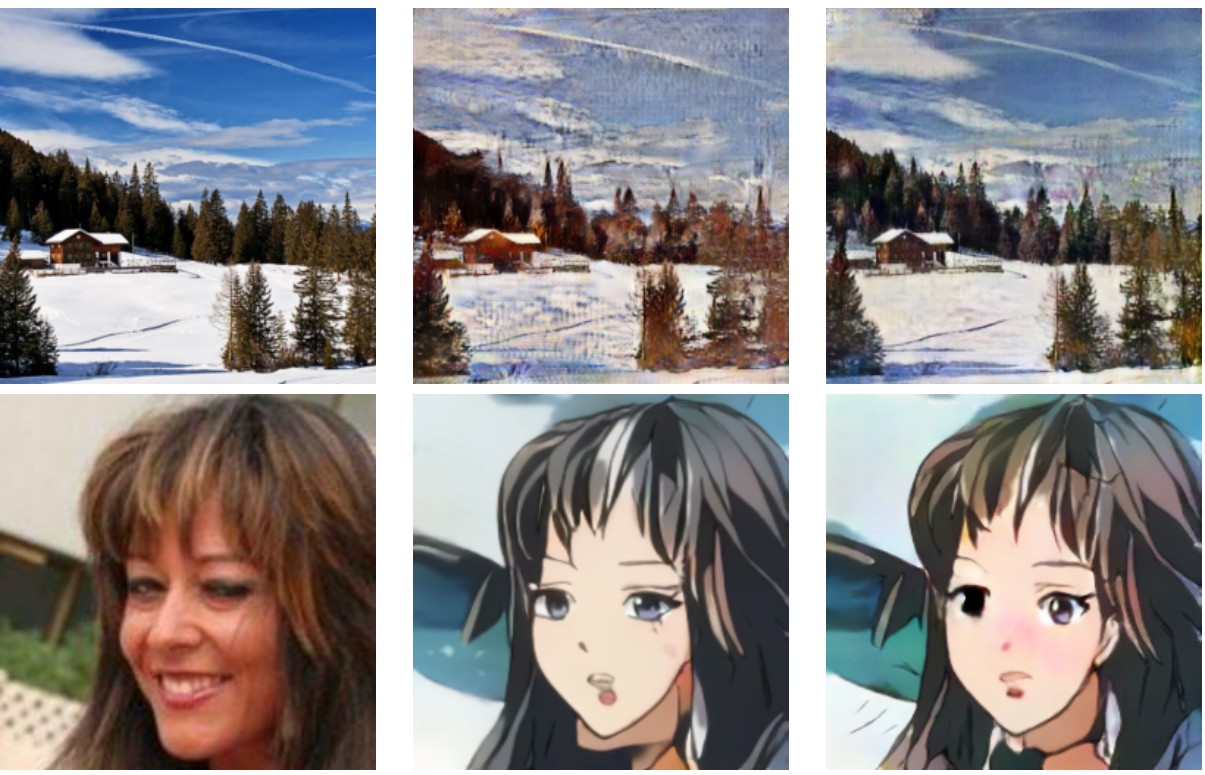

Figure 1: Example comparison of style transfer. **left**: original, **center**: $F_\mathcal{V}$ output, **right**: $F_\mathcal{A}$ output. Top row: Monet-to-Photo style transfer, bottom row: Selfie-to-Anime style transfer. Images are down-scaled to fit the page.

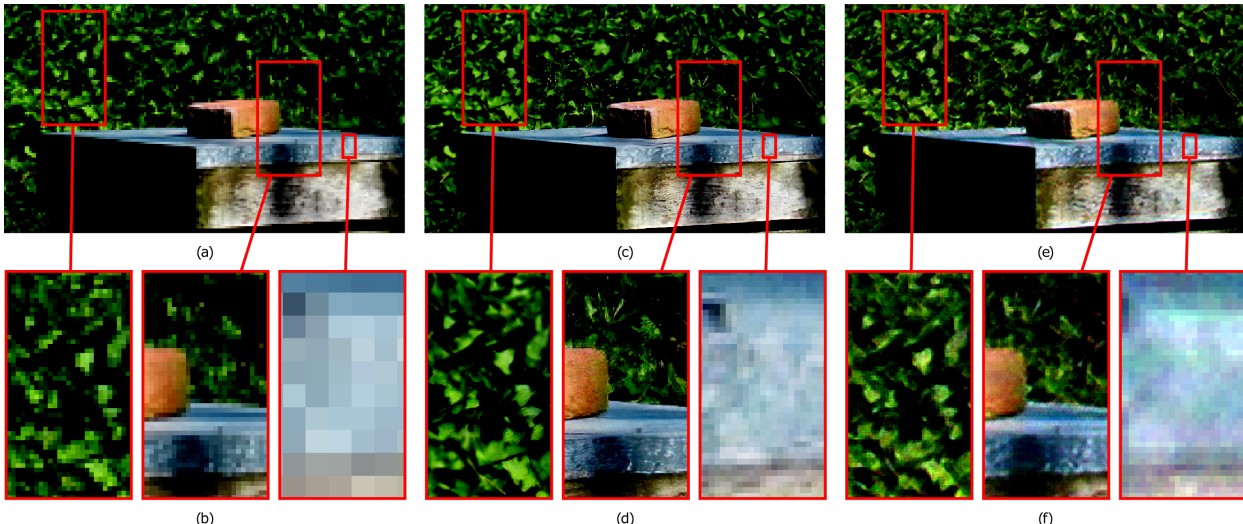

Figure 2: Example comparison of a Super-Resolution transformation. **Left**: original (a), **center**: $F_\mathcal{V}$ output (c), **right**: $F_\mathcal{A}$ output (e). Image up-scaled x4 from 175x100. We zoom in selected parts of the images (b,d,f) to show the difference in resolution among the three images.

In this paper, we show that an adversary can successfully build surrogate models mimicking the functionality of several types of image translation models (Figures 1-2). Our attack does not require the adversary to have access to victim's training data, knowledge of victim model parameters or its architecture. We make the following contributions:

1. present the **first model extraction attack against image translation models** (Section 4), and

2. empirically demonstrate its effectiveness via: **1) quantitative analyses** using perceptual similarity metrics on three datasets spanning two types of image translation tasks: style transfer and image super resolution; the differences between the victim and surrogate models, compared to the target, are in the following ranges – Selfie-to-Anime: Frechet Information Distance (FID) $13.36 - 39.66$, Monet-to-Photo[1]: FID $3.57 - 4.40$, and Super-Resolution: Structural Similarity: $0.06 - 0.08$ and Peak Signal-to-Noise Ratio: $1.43 - 4.46$ (Section 5); **2) an extensive user study** ($n = 125$) on the style transfer tasks (Selfie-to-Anime and Monet-to-Photo) showing that human perception of images produced by the victim and surrogate models are similar, within an equivalence bound of Cohen's $d = 0.3$. (Section 5.4); and

3. highlight the need for novel defenses by showing that **existing defenses against model extraction are ineffective** for image translation models (Section 7).

## 2 Background and Related Work

In this section, we introduce the background necessary to understand the rest of this paper. Also, we give an overview of the use of generative adversarial networks (GANs) for image translation tasks, and a set of standardized metrics used to measure the effectiveness of such translations.

### 2.1 Model Extraction Attacks and Defenses

The goal of the adversary $\mathcal{A}$ in model extraction attacks is to build a surrogate model to mimic the functionality of a victim model (Orekondy et al., 2019; Papernot et al., 2017). The motivations for model extraction can be manifold: to use the surrogate model to offer a competing service (Correia-Silva et al., 2018; Orekondy et al., 2019), or to probe the surrogate model to find adversarial examples that transfer to the victim model itself (Juuti et al., 2019; Papernot et al., 2017). $\mathcal{A}$ conducts the attack by repeatedly querying the inference interface of the victim and using the responses to train the surrogate.

The techniques used for model extraction can differ depending on the assumptions regarding the knowledge and capabilities of the adversary. It is typical to assume that $\mathcal{A}$ does not have access to the victim's training data but may know the general domain (e.g., pictures of animals) (Papernot et al., 2017; Correia-Silva et al., 2018). It is also customary to assume that $\mathcal{A}$ does not know the exact architecture of the victim model. However, $\mathcal{A}$ may infer information about the victim architecture using side-channel attacks (Batina et al., 2019; Hua et al., 2018; Duddu et al., 2019), snooping on shared caches (Yan et al., 2020), or matching model's responses to a series of queries with known architectures (Oh et al., 2018).

Previous work on model extraction attacks focuses primarily on image classification tasks (Tramèr et al., 2016; Papernot et al., 2017; Juuti et al., 2019). Examples include attacks against ImageNet-scale models where $\mathcal{A}$'s goal was *functionality stealing* (Orekondy et al., 2019; Correia-Silva et al., 2018; Jagielski et al., 2020). Other than image classification, there is a large body of work targeting other types of models - NLP models (Krishna et al., 2020; Wallace et al., 2021), GNNs (He et al., 2020), and RNNs (Takemura et al., 2020). Existing attack against GANs (Hu & Pang, 2021) is limited to models that generate images from latent codes, and is not applicable to image translation models.

Existing model extraction defenses either attempt to identify $\mathcal{A}$ by studying the query distribution (Juuti et al., 2019; Atli et al., 2020) or try to identify queries that are close to decision boundaries (Quiring et al., 2018; Zheng et al., 2019), or explore an abnormally large part of the hyperspace (Kesarwani et al., 2018). Alternatively, the defender can perturb the prediction vector before sending it back to the client to slow down the construction of effective surrogate models (Kariyappa & Qureshi, 2019; Orekondy et al., 2020).

Instead of detecting or defending against model extraction attacks, the defender can watermark their model and try to claim ownership if $\mathcal{A}$ makes their surrogate model public. Watermarks can be embedded into the model during training by modifying its loss function (Jia et al., 2021) that makes it more likely that the watermarks transfer to the surrogate model. Alternately, model responses can be modified at the API level to embed a client-specific watermark

---

[1]The task is translating from photos to Monet paintings, but we refer to it as "Monet-to-Photo" to be consistent with the name of the dataset (Zhu et al., 2017).

into $\mathcal{A}$'s training data (Szyller et al., 2020). Both schemes are based on the idea that the watermark must extend to *all* models *derived from* $F_\mathcal{V}$, and not just the $F_\mathcal{V}$. Watermarking schemes that do not consider this were shown not be resilient to model extraction attacks (Shafieinejad et al., 2019).

While there do exist watermarking schemes for GANs, they either primarily assume a black-box generator style of GAN (Yu et al., 2021) or that $\mathcal{A}$ is going to use specific data to launch the attack (Zhang et al., 2020).

## 2.2 Generative Adversarial Networks

Generative Adversarial Networks (GANs) are generative models which learn the underlying data distribution as a game between two models: a generator model $G$ which generates new images from input noise and a discriminator model $D$ which attempts to distinguish real images from the training data and fake images from $G$ (Goodfellow et al., 2014). This is modeled as a minimax optimization problem: $\min_{q_\phi} \max_D V(G, D)$, where $D$ maximises its gain $V$ obtained by correctly distinguishing real and fake generated images. $G$ minimizes the maximum gain by $D$ and iteratively improves the quality of images to be close to the original training data points while fooling $D$. The overall optimization objective can be formulated as follows:

$$
\begin{aligned}
V(G, D) = & \; \mathbb{E}_{x \sim p_{data}(x)}[log(D(x))] \\
& + \mathbb{E}_{z \sim p_z(z)}[1 - log(D(G(z)))]
\end{aligned}
\tag{1}
$$

where $p_{data}$ is the distribution of the training data and $p_z$ is the distribution learnt by $G$. This optimization is solved by alternatively training $D$ and $G$ using some variant of gradient descent, e.g. SGD or Adam. The outstanding performance of GANs have resulted in their widespread adoption for different applications addressing several image processing and image translation tasks. In this work, we consider two main applications of image to image translations: style transfer (Zhu et al., 2017; Isola et al., 2017; Kim et al., 2020) and image super resolution (Ledig et al., 2017).

**Style Transfer.** GANs have been used to translate images from one style to another (Isola et al., 2017). Prior work explored several image style transfer tasks such as changing seasons, day to night, black and white to colored, and aerial street images to maps (Isola et al., 2017; Zhu et al., 2017; Kim et al., 2020). Some specific tasks have been extensively adopted as filters in social networks such as Instagram, Snapchat and FaceApp, e.g. changing human faces to anime characters (Kim et al., 2020), aging or swapping genders (Karras et al., 2019; Choi et al., 2018). These are trained using GANs of different types. Style transfer GANs can be trained using paired image-to-image data with a Pix2Pix model (Isola et al., 2017) in a supervised fashion - each unstyled image has its styled counterpart. For complex tasks, such supervised image to image translation training data is difficult, if not impossible, to obtain. To address this, training GANs using unpaired and uncorrelated mappings of the images in the input source domain to the target output domain has been proposed (e.g. CycleGAN, UGATIT (Kim et al., 2020; Zhu et al., 2017)).

*Pix2Pix* (Isola et al., 2017) architecture uses a UNet Generator Model which downsamples an input image from the source domain and outputs an upsampled image in the target domain. It uses a PatchGAN discriminator network which takes the input image from the source domain and the image from the target domain to determine whether the generated output image from the target domain is from the real distribution or from the UNet. To train a Pix2Pix model, the UNet generator is trained to minimize the loss of correctly translating the input source domain image to the target domain image ($IdentityLoss$) while additionally minimizing the adversarial loss that captures the success of discriminator, to produce realistic target domain images. The overall loss optimized includes: $GeneratorLoss = AdversarialLoss + \lambda * IdentityLoss$.

*CycleGAN* (Zhu et al., 2017) consists of two GANs. One of the GANs maps images from the source domain to the target domain while the second GAN maps images from the target domain back to the source domain. Both generators are similar to generators in Pix2Pix and map images from one domain to the other by downsampling the image, followed by upsampling the image to the target domain. Like Pix2Pix, CycleGAN uses adversarial loss with respect to the discriminators to ensure that the generator outputs domain images close to real images, and Identity Loss to ensure similarity between the generated image and real target domain image. Additionally, CycleGAN uses the optimized cycle consistency loss. It ensures that an image when mapped from the source domain to the target domain and back to the source domain are as similar as possible.

*UGATIT* (Kim et al., 2020) utilizes convolutional neural networks as an attention module to identify and extract major features in images, in addition to the architecture and optimization of the CycleGAN. This allows for finer optimization while translating input images to the target domain.

**Image Super Resolution.** Maps low resolution images to high resolution images using a GAN. The generator includes residual blocks that alleviate the vanishing gradient problem and hence, enable deeper models that give better results. The generator outputs a high resolution (HR) image that is an up-scaled version of its low resolution input counterpart. The discriminator differentiates between the generated HR image with the ground truth HR image. The generator parameters are updated similarly to a generic GAN (Equation 1).

### 2.3 Metrics

We consider several metrics for comparing the quality of images generated by the victim and surrogate models. Unlike classification models that rely on accuracy with respect to ground truth, we need to be able to objectively compare the performance of the victim and surrogate models without necessarily having access to any ground truth.

**Structural Similarity (SSIM)** (Zhou Wang et al., 2004) of two image windows $x$ and $y$ of same size $N \times N$, is:

$$SSIM(x,y) = \frac{(2\mu_x\mu_y + c_1)(2\sigma_{xy} + c_2)}{(\mu_x^2 + \mu_y^2 + c_1)(\sigma_x^2 + \sigma_y^2 + c_2)} \tag{2}$$

where $\mu$ is the mean of pixels over the windows, $\sigma^2$ is the variance of pixel values and $\sigma_{xy}$ is the covariance, $c_1$ and $c_2$ are constants to ensure numerical stability. SSIM compares two images with one of the images as a reference. Dissimilar images are scored as $0.0$ while the same images are given a score of $1.0$.

**Peak Signal to Noise Ratio (PSNR)** (Horé & Ziou, 2010) is the ratio of the signal (image content) and the noise that lowers the fidelity of the image:

$$PSNR = 20.log_{10}\frac{MAX}{\sqrt{MSE}} \tag{3}$$

where MAX is maximum range of pixel value in the image given by $2^b - 1$ where b is the number of bits. MSE is the mean squared error which is the squared difference between the image pixel values and the mean value.

We use SSIM and PSNR to compare the quality of the generated super resolution image to the original HR ground truth image. For the style transfer tasks, we consider Frechet Inception Distance instead.

**Frechet Inception Distance (FID)** measures the distance between the features of real and generated images, and is used to estimate the quality of images from GANs (Heusel et al., 2017; Lucic et al., 2018). FID uses the features from the last pooling layer before the output prediction layer from the Inceptionv3 network. The distance between the feature distributions for two sets of images is computed using Wasserstein (Frechet) distance. Low scores correspond to higher quality generated images but the magnitude of what constitutes high or low FID depends on the domain and complexity of measured images.

## 3 Problem Statement

**Adversary Model.** The adversary $\mathcal{A}$ wants to build a surrogate model $F_{\mathcal{A}}$ to mimic the functionality of a model $F_{\mathcal{V}}$ belonging to the victim $\mathcal{V}$. $\mathcal{A}$ is aware of the purpose of $F_{\mathcal{V}}$, e.g. style transfer from Monet paintings, and can choose an appropriate model architecture of $F_{\mathcal{A}}$. However, $\mathcal{A}$ is not aware of the exact architecture of $F_{\mathcal{V}}$. $\mathcal{A}$ conducts the attack with any data $X_{\mathcal{A}}$ they want but they have no knowledge of the exact training data $\mathcal{V}$ used. Crucially, we assume that $\mathcal{A}$ does not have access to any *source style* images $S_X$ that contain the style template $S$. We assume that $\mathcal{V}$ wants to keep $S_X$ secret because it is valuable and difficult to obtain. $\mathcal{A}$ has only *black-box* access to $F_{\mathcal{V}}$. $\mathcal{A}$ uses $X_{\mathcal{A}}$ and $X_{\mathcal{A}_S} = F_{\mathcal{V}}(X_{\mathcal{A}})$ to train $F_{\mathcal{A}}$ in any way they want. $\mathcal{A}$'s goal is to duplicate the *functionality* of $F_{\mathcal{V}}$ as closely as possible.

Table 1: Summary of the notation used throughout this work.

| | | | |
|---|---|---|---|
| $\mathcal{V}$ | victim | $X$ | unstyled image(s) |
| $\mathcal{A}$ | adversary | $X_S$ | styled image(s) |
| $F_{\mathcal{V}}$ | victim model | $X_{\mathcal{A}}$ | $\mathcal{A}$'s unstyled image(s) |
| $F_{\mathcal{A}}$ | surrogate model | $X_{\mathcal{A}_S}$ | $\mathcal{A}$'s styled images |
| $S$ | style template | $X_{lr}/X_{hr}$ | low/high resolution image(s) |
| $S_X$ | source style images | $X_{\mathcal{A}_{lr}}/X_{\mathcal{A}_{hr}}$ | $\mathcal{A}$'s low/high resolution image(s) |

**Goals.** $\mathcal{A}$'s goal is not to optimize the style transfer task itself but rather to achieve a level of performance comparable to $F_{\mathcal{V}}$. Therefore, they use $F_{\mathcal{V}}$ as an oracle to obtain its *"ground truth"* for training $F_{\mathcal{A}}$. $\mathcal{A}$ wants to either keep using the service without paying or to offer a competitive service at a discounted price. $\mathcal{A}$ is not trying to obtain the exact replica of $F_{\mathcal{V}}$. Upon a successful model extraction attack, images styled by $F_{\mathcal{A}}$ should be comparable to those produced by $F_{\mathcal{V}}$. Assuming $F_{\mathcal{V}}$ is a high quality style-transfer model, then the attack is considered successful if $F_{\mathcal{V}}(X) \approx F_{\mathcal{A}}(X)$.

## 4 Extracting Image Translation Models

We target two most popular applications of GANs for image translation: image style transfer (Selfie-to-Anime and Monet-to-Photo) and image super resolution. The attack setting considers $\mathcal{V}$'s model $F_{\mathcal{V}}$: $X \to X_S$ that maps an unstyled original image $X$ to the corresponding styled image $X_S$. The style template $S$ is added to the input image. $\mathcal{A}$ aims to steal the styling functionality of $F_{\mathcal{V}}$ that maps the input image to the corresponding styled image without explicit access to the secret styling template $S$. We summarise the notation in Table 1.

### 4.1 Attack Methodology

$\mathcal{A}$ queries $F_{\mathcal{V}}$ using $\mathcal{V}$'s API to obtain the corresponding styled $X_{\mathcal{A}_S}$ images generated by $F_{\mathcal{V}}$. This in consistent with the practical settings likely to be encountered in real world applications. This pair of unstyled input and styled output image $(X_{\mathcal{A}}, X_{\mathcal{A}_S})$ constitutes the surrogate training data used by $\mathcal{A}$ to train the surrogate model $F_{\mathcal{A}}$. Since $\mathcal{A}$ relies on training $F_{\mathcal{A}}$ using pairs of data points, we use paired image translation models as our attack model. As previously explained, training paired translation models is not realistic in many real world applications due to the lack of styled images. However in our case, $\mathcal{A}$ leverages the fact that they can obtain many $X_{\mathcal{A}_S}$ by querying $F_{\mathcal{V}}$. Additionally, our attack makes no assumptions regarding the styling technique used to generate the styled images. However, models that we target do require GANs for high quality transformations. Even though the styling template $S$ applied to unstyled input images by $F_{\mathcal{V}}$ is learned explicitly from images, it can be stolen and implicitly embedded in $F_{\mathcal{A}}$.

More formally, from the set of possible mappings $\mathcal{F}$, the mapping $F_{\mathcal{V}} : X \to X_S$ is optimal if according to some quantitative criterion $\mathcal{M}_S$, the resulting transformation captures the style $S$ by maximizing $\mathcal{M}_S(X_S, S)$:

$$F_{\mathcal{V}} = arg \max_{F'_{\mathcal{V}} \in \mathcal{F}} \mathcal{M}_S(X_S, S) \tag{4}$$

as well as retains the semantics of the input image $\mathcal{M}_X(X, X_S)$ by maximizing another criterion $\mathcal{M}_X$:

$$F_{\mathcal{V}} = arg \max_{F'_{\mathcal{V}} \in \mathcal{F}} \mathcal{M}_X(X, X_S) \tag{5}$$

$\mathcal{A}$'s goal is to learn the same kind of mapping $F_{\mathcal{A}} : X_{\mathcal{A}} \to X_{\mathcal{A}_S}$ using images obtained from $F_{\mathcal{V}}$. This transformation must be successful according to both criteria $\mathcal{M}_X(X_{\mathcal{A}}, X_{\mathcal{A}_S})$ and $\mathcal{M}_S(X_{\mathcal{A}_S}, S)$:

$$F_{\mathcal{A}} = arg \max_{F'_{\mathcal{A}} \in \mathcal{F}} \mathcal{M}_X(X_{\mathcal{A}}, X_{\mathcal{A}_S}) \quad and$$

$$F_{\mathcal{A}} = arg \max_{F'_{\mathcal{A}} \in \mathcal{F}} \mathcal{M}_S(X_{\mathcal{A}_S}, S) \tag{6}$$

However, $\mathcal{A}$ does not have access to $S_X$, and thus the template $S$. Instead, they use a proxy metric $\mathcal{M}_\mathcal{P}(F_\mathcal{V}, F_\mathcal{A}, X)$. Abusing notation, we define it as:

$$F_\mathcal{A} = arg \max_{F'_\mathcal{A} \in \mathcal{F}} M_\mathcal{P}(F_\mathcal{V}, F_\mathcal{A}, X) \quad s.t.$$

$$if \quad F_\mathcal{A}(X) \approx F_\mathcal{V}(X) \quad then$$

$$F_\mathcal{A} = arg \max_{F'_\mathcal{A} \in \mathcal{F}} \mathcal{M}_X(X_\mathcal{A}, X_{\mathcal{A}_S}) \quad and$$

$$F_\mathcal{A} = arg \max_{F'_\mathcal{A} \in \mathcal{F}} \mathcal{M}_S(X_{\mathcal{A}_S}, S) \tag{7}$$

In other words, we assume that if images produced by $F_\mathcal{A}$ are similar to those produced by $F_\mathcal{V}$, then $F_\mathcal{A}$ has successfully learned the style $S$.

We apply our attacks to the following three settings.

**Setting 1: Monet-to-Photo.** Monet-to-Photo is a style transfer task in which a picture of a scene is transformed to look as if it was painted in Claude Monet's signature style. We train $F_\mathcal{V}$ as an unpaired image to image translation problem, i.e, the training data consists of unordered and uncorrelated photos of landscapes, and Monet's paintings $S_X$ that contains his signature style $S$. $S_X$ is secret and not visible to $\mathcal{A}$. The model learns the features in the Monet paintings and applies them to translate generic photos. $\mathcal{A}$ queries $F_\mathcal{V}$ with images $X_\mathcal{A}$ from a different dataset than $\mathcal{V}$'s training data and obtains corresponding $X_{\mathcal{A}_S} = F_\mathcal{V}(X_\mathcal{A})$. $F_\mathcal{A}$ is trained using this generated dataset $(X_\mathcal{A}, X_{\mathcal{A}_S})$ as a paired image to image translation problem between the two domains.

**Setting 2: Selfie-to-Anime.** Selfie-to-Anime is used as part of a real world web application which allows users to upload their selfie and convert it to a corresponding anime image (Rico Beti). We do not attack the actual web interface due to its terms of service on reverse engineering and scraping. However, the authors disclose the data and architecture (Kim et al., 2020) used to train their model, and we use it to train our $F_\mathcal{V}$. The training data is unpaired - it consists of a training set of selfie images, and a styling template $S$ consisting of anime images $S_X$. $\mathcal{A}$ queries $F_\mathcal{V}$ with images $X_\mathcal{A}$ from a different dataset than $\mathcal{V}$'s training data and obtains corresponding $X_{\mathcal{A}_S} = F_\mathcal{V}(X_\mathcal{A})$. $F_\mathcal{A}$ is trained using this generated dataset $(X_\mathcal{A}, X_{\mathcal{A}_S})$ as a paired image to image translation problem between the two domains.

**Setting 3: Super-Resolution.** In Super-Resolution a low resolution image is upscaled to its high resolution counterpart using a GAN. We consider mapping low resolution images, (using a 4x bicubic downscaling) to their high resolution counterparts with 4x upscaling. $F_\mathcal{V}$'s generator is trained in a supervised fashion using a paired dataset $(X_{lr}, X_{hr})$ of low resolution and corresponding high resolution images. Here $X_{hr}$ corresponds to $X_S$. There is no explicit styling template $S$. $\mathcal{A}$ queries $F_\mathcal{V}$ with publicly available low resolution images $X_{\mathcal{A}_{lr}}$ to obtain the corresponding high resolution image $X_{\mathcal{A}_{hr}} = F_\mathcal{V}(X_{\mathcal{A}_{lr}})$. This paired surrogate data $(X_{\mathcal{A}_{lr}}, X_{\mathcal{A}_{hr}})$ is used to train $F_\mathcal{A}$.

Our attack does not make any assumptions about the training or the architecture of $F_\mathcal{V}$, and it relies solely on the query access. Hence, it is applicable to other image translation settings and can be used to extract $F_\mathcal{V}$s with a well-defined style $S$.

# 5 Evaluation

## 5.1 Datasets

**Monet-to-Photo.** $F_\mathcal{V}$ is trained using the Monet2Photo (Mon) dataset consisting of 1193 Monet paintings and 7038 natural photos divided into train and test sets. For training $F_\mathcal{A}$, we use the Intel Image Classification dataset (int). It consists of 14,034 of images of various nature and city landscapes. As the benchmark dataset, we choose the Landscape (lan) dataset that contains 13,233 images from the same domain but was not used to train either model.

**Selfie-to-Anime.** $F_\mathcal{V}$ is trained on the selfie2anime dataset (sel) which comprises 3400 selfies and 3400 anime faces as part of the training data, and 100 selfies and 100 anime faces in the test set. $\mathcal{A}$ uses the CelebA (Liu et al., 2015)

Table 2: Datasets used to for training $F_{\mathcal{V}}$ and $F_{\mathcal{A}}$.

| Task | $\mathcal{V}$ Data | | $\mathcal{A}$ Data | Benchmark Data |
|------|------|------|------|------|
| Monet-to-Photo | Monet2Photo | | Intel | Landscape |
| | Train | Test | Train | Test |
| | 6287 | 751 | 14034 | 4319 |
| Selfie-to-Anime | Selfie2Anime | | CelebA | LFW |
| | Train | Test | Train | Test |
| | 3400 | 100 | 162770 | 13233 |
| Super-Resolution | DIV2K | | FLICKR2K | SRBenchmark |
| | Train | Test | Train | Test |
| | 800 | 200 | 2650 | 294 |

dataset for attacking $\mathcal{V}$ which consists of 162,770 photos of celebrities. For the benchmark dataset, we use the LFW dataset (Huang et al., 2007) which contains a test set of 13233 selfie images.

**Super-Resolution.** $F_{\mathcal{V}}$ is trained using the DIV2K (Agustsson & Timofte, 2017) dataset that consists of 800 pairs of low and corresponding high resolution images in the training set, and 200 image pairs for validation and testing. $\mathcal{A}$ uses the FLICKR2K dataset for the extraction. It consists of 2650 train (and 294 test images). We create the SRBechmark dataset by combining multiple benchmarking datasets used for super resolution model evaluations including SET5 (Lai et al., 2017), SET14 (Lai et al., 2017), URBAN100 (Lai et al., 2017) and BSD100 (Lai et al., 2017) containing 294 images. All high resolution images are 4x bicubic downsampled.

## 5.2 Architectures

For **Monet-to-Photo**, we use the state of the art CycleGAN (Zhu et al., 2017) architecture as the $F_{\mathcal{V}}$ architecture. For $F_{\mathcal{A}}$ architecture we use a paired image-to-image translation model Pix2Pix (Isola et al., 2017). For **Selfie-to-Anime**, we use the UGATIT generative model (Kim et al., 2020) as the $F_{\mathcal{V}}$ architecture which is similar to a CycleGAN. This is the model used by the Selfie2Anime service (Rico Beti). Like for Monet-to-Photo, we use a paired image-to-image translation model Pix2Pix (Isola et al., 2017) for $F_{\mathcal{A}}$. For **Super-Resolution**, we use the state of the art SRGAN model (Ledig et al., 2017) or the $F_{\mathcal{V}}$ architecture, and for $F_{\mathcal{A}}$ we use a similar SRResNet model (Ledig et al., 2017) with ResNet convolutional units.

## 5.3 Experiments

Tables 3 to 5 summarize the results of our experiments. We report the mean value for a given distance metric, and standard deviations where applicable. In Tables 3 and 4 we report the effectiveness of the attack measured using the FID score for style transfer settings. In Table 5 we report the effectiveness of the attack measured using PSNR and SSIM for the Super-Resolution setting. In each setting for each dataset, we report on three different experiments:

(A) **effectiveness of $F_{\mathcal{V}}$ transformations:** this is the baseline performance of $F_{\mathcal{V}}$ with respect to the original test set (cell **(A)** in Tables 3 to 5).

(B) **comparison of $F_{\mathcal{A}}$ and $F_{\mathcal{V}}$:** this is measured using the proxy metric (Equation 7) that $\mathcal{A}$ uses to assess the effectiveness of $F_{\mathcal{A}}$ in terms of how closely its outputs resemble the outputs of $F_{\mathcal{V}}$ (cell **(B)** in Tables 3 to 5).

(C) **effectiveness of $F_{\mathcal{A}}$ transformations:** this is measured the same way as **(A)**. Note that $\mathcal{A}$ cannot directly compute this effectiveness metric because they do not have access to the ground truth (in the form of the styling set $S$ or paired images in the case of Super-Resolution; cell **(C)** in Tables 3 to 5).

For Monet-to-Photo, we can see that our attack is effective at extracting $F_{\mathcal{V}}$. In Table 3, we report the FID scores for the three experiments using two test sets. In all cases, we see that $F_{\mathcal{A}}$ performs comparably to $F_{\mathcal{V}}$.

Our attack is effective at extracting a Selfie-to-Anime $F_{\mathcal{V}}$. In Table 4, we report the FID scores for the three experiments, using two test sets. We observe a large FID score for the $\mathcal{V}$'s test set (Selfie2Anime). However, using a

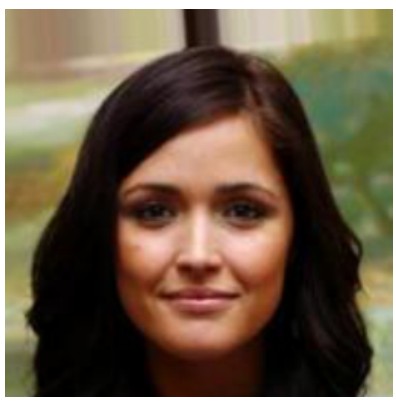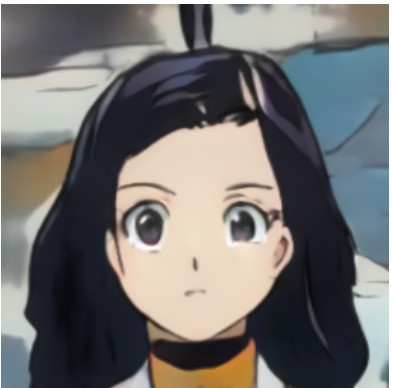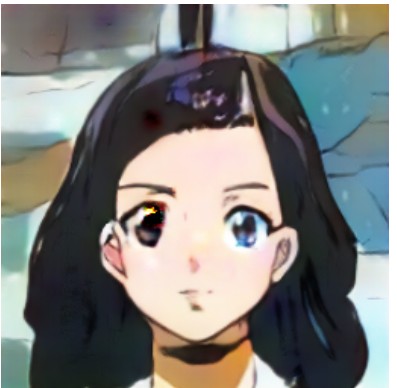

Figure 3: Example of successful transformations produced by $F_{\mathcal{V}}$ and $F_{\mathcal{A}}$ that are visually similar.

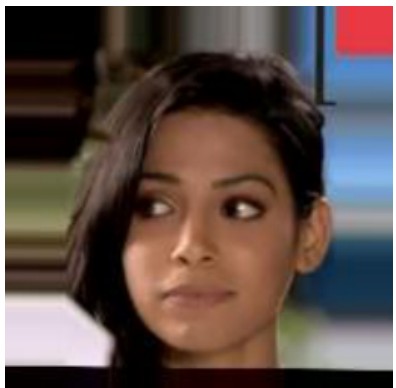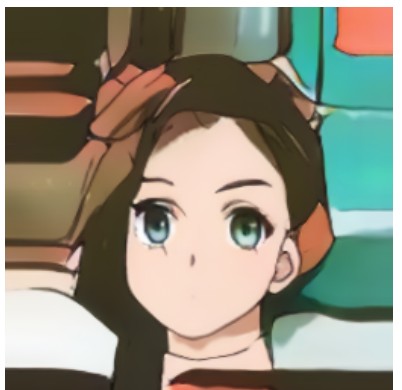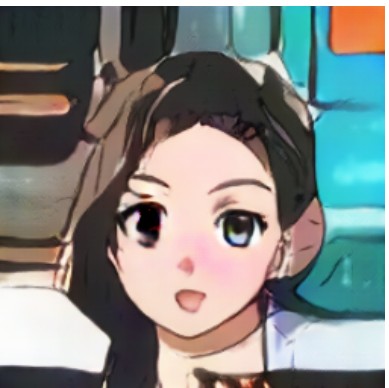

Figure 4: $F_{\mathcal{V}}$ and $F_{\mathcal{A}}$ learn different representations and can produce transformations that have different features even though both transformations are comparably effective.

benchmark test set (LFW), we observe that both models compare similarly and produce similar styled images, c.f. Figure 3. Additionally, we observe that many transformations are qualitatively similar (both models successfully produce anime faces) even though the resulting transformations are different, c.f. Figure 4.

Finally, our attack is effective at extracting the Super-Resolution $F_{\mathcal{V}}$, c.f. Table 5. Both in the case of SSIM (Table 5a) and PSNR (Table 5b), $F_{\mathcal{A}}$ performs similarly to $F_{\mathcal{V}}$. This holds for the proxy comparison between $F_{\mathcal{V}}$ and $F_{\mathcal{A}}$, as well as w.r.t. the original high resolution images.

## 5.4 User Study

Since distance metrics do not necessarily capture human perception, we conducted an extensive user study to evaluate how humans rate the image translation tasks. All user study materials are available in the supplement.

**Materials.** We designed a questionnaire in Google Forms which we shared with the participants.

**Participants.** We conducted the user study with 125 participants invited through social media (Twitter, LinkedIn, Facebook) and institutional channels (Slack, Teams, Telegram groups). We collected basic demographics about the participants, including age, highest obtained education, and gender. Our user study data collection conforms to our institution's data management guidelines. No personally identifiable information was collected as part of the study. Figure 5 represents the demographics of the participants of the study.

**Design of User Study.** In the questionnaire, we first introduced the problem of image style transfer with examples of Monet paintings and anime images. We explained that the study aims to evaluate which of the shown style transformations are successful in the participant's opinion, and does not evaluate their skill in making this judgement. Participants

Table 3: FID of $F_V$ and $F_A$ for Monet-to-Photo. We report the mean and standard deviation over five runs.

| Output Distribution | Test Set | Compared Distribution | | | |
|---|---|---|---|---|---|
| | | $V$ Output | | Monet Paintings | |
| $V$ Output | Monet2Photo Test | N/A | (A) | $58.05 \pm 0.74$ | |
| | Landscape | N/A | | $49.59 \pm 0.82$ | |
| $A$ Output | Monet2Photo Test | (B) | $42.86 \pm 1.01$ | (C) | $61.62 \pm 1.05$ |
| | Landscape | | $15.38 \pm 0.54$ | | $53.99 \pm 1.21$ |

Table 4: FID of $F_V$ and $F_A$ for Selfie-to-Anime. We report the mean and standard deviation over five runs.

| Output Distribution | Test Set | Compared Distribution | | | |
|---|---|---|---|---|---|
| | | $V$ Output | | Anime Faces | |
| $V$ Output | Selfie2Anime | N/A | (A) | $69.02 \pm 1.91$ | |
| | LFW Test | N/A | | $56.06 \pm 2.47$ | |
| $A$ Output | Selfie2Anime | (B) | $106.63 \pm 5.53$ | (C) | $108.68 \pm 3.19$ |
| | LFW Test | | $19.67 \pm 0.69$ | | $69.42 \pm 2.04$ |

then evaluated the style transformations using a Likert scale from 1 to 5. A score of 5 indicated that the result looked very similar to a Monet painting of the input scene or an anime character corresponding to the source selfie photo, and a score of 1 that there was no such resemblance.

**Procedure.** We conducted the user study on two style transfer settings. For both tasks, we included 20 pairs consisting of an original unstyled image and its corresponding styled variant. Half of the them (10) were styled with $F_V$ and the other half with $F_A$. The pairs were shown in random order.

**Results.** Figure 6 shows the distribution of scores for $F_V$ and $F_A$ in the Monet-to-Photo and Selfie-to-Anime test settings, respectively. Our evaluation concerns whether the differences between results obtained from $F_V$ and $F_A$ were *statistically significant*. We first conducted a *t-test*, which yields a conditional probability (*p-value*) for receiving values deviating equally or more between the sample statistics as observed in the experiments, under the *null hypothesis* of equal underlying population distributions. The outcome is considered statistically significant if $p$ exceeds the *significance threshold* $\alpha$, which we set to $0.05$. Since sample variances were unequal between $F_V$ and $F_A$, we used Welch's t-test instead of Student's t-test.

The null hypothesis of equivalent population distributions was rejected for Monet-to-Photo ($F_V$'s score mean $= 3.20$ and std $= 1.59$; $F_A$'s score mean $= 2.91$ and std $= 1.76$; $p = 1.8e - 8$) but not for Selfie-to-Anime ($F_V$'s score mean $= 3.11$ and std $= 1.76$; $F_A$'s score mean $= 3.08$ and std $= 1.50$; $p = 0.63$). In other words, the probability of obtaining results deviating between $F_V$ and $F_A$ as much as observed (or more) would be very high for Selfie-to-Anime but very low for Monet-to-Photo, if $F_V$ and $F_A$ indeed performed similarly.

However, the t-test alone is insufficient to evaluate the divergence between $F_V$ and $F_A$. We additionally performed a *two one-sided t-test* (TOST) for *equivalence* to test whether the results fall between *equivalence bounds*. As the bounds, we chose the range $[-0.3, 0.3]$ for *Cohen's d*, which is the mean difference between the models' scores standardized by their pooled standard deviations (Cohen, 1988). The null hypothesis is reversed from the standard t-test, now assuming the *non-equivalence* between underlying population distributions. The null hypothesis is rejected if the observed difference falls within the equivalence bounds.

For Selfie-to-Anime the null hypothesis was rejected within the equivalence bounds of $[-0.11, 0.11]$ ($p = 0.0476$), and for Monet-to-Photo within $[-0.38, 0.38]$ ($p = 0.04829$), using $\alpha = 0.05$. Both fall within $[-0.3, 0.3]$ for Cohen's $d$, which corresponds to the raw value range of approximately $[-0.50, 0.50]$ in Selfie-to-Anime and $[-0.49, 0.49]$ in Monet-to-Photo. Hence, we reject the non-equivalence of $F_V$ and $F_A$ in **both** test settings.

While a statistically significant difference between $F_V$ and $F_A$ was observed in Monet-to-Photo, it was too small to count as sufficiently large for meaningful non-equivalence in TOST. In Selfie-to-Anime, both the t-test and TOST supported rejecting the non-equivalence of $F_V$ and $F_A$. We therefore conclude that $F_A$ **successfully achieved performance close to $F_V$ based on human judgement**.

Table 5: PSRN and SSIM of $F_\mathcal{V}$ and $F_\mathcal{A}$ for Super-Resolution. We report the mean and standard deviation over five runs.

(a) **SSIM**. Comparing $F_\mathcal{A}$ to $F_\mathcal{V}$ captures the true performance.

| Output Distribution | Test Set | Compared Distribution | | | |
| --- | --- | --- | --- | --- | --- |
| | | | $\mathcal{V}$ Output | | Original high-res |
| $\mathcal{V}$ Output | DIV2K Test | | N/A | **(A)** | $0.74 \pm 0.05$ |
| | SRBENCHMARK | | N/A | | $0.69 \pm 0.12$ |
| $\mathcal{A}$ Output | DIV2K Test | **(B)** | $0.75 \pm 0.08$ | **(C)** | $0.80 \pm 0.10$ |
| | SRBENCHMARK | | $0.76 \pm 0.09$ | | $0.61 \pm 0.06$ |

(b) **PSNR**. Comparing $F_\mathcal{A}$ to $F_\mathcal{V}$ captures the true performance.

| Output Distribution | Test Set | Compared Distribution | | | |
| --- | --- | --- | --- | --- | --- |
| | | | $\mathcal{V}$ Output | | Original high-res |
| $\mathcal{V}$ Output | DIV2K Test | | N/A | **(A)** | $24.67 \pm 1.81$ |
| | SRBENCHMARK | | N/A | | $22.59 \pm 4.51$ |
| $\mathcal{A}$ Output | DIV2K Test | **(B)** | $24.74 \pm 5.04$ | **(C)** | $23.24 \pm 3.77$ |
| | SRBENCHMARK | | $20.64 \pm 3.33$ | | $18.13 \pm 3.61$ |

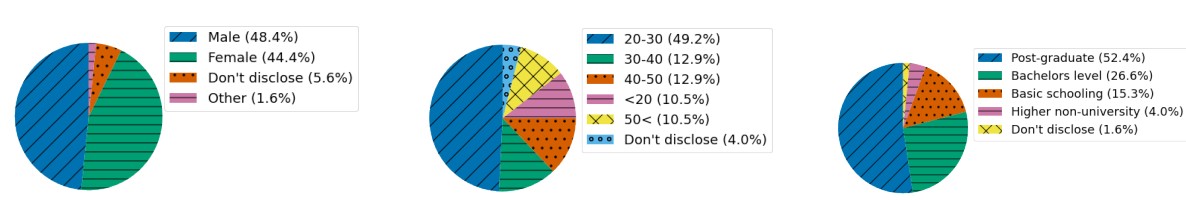

Figure 5: Distribution of participant demographics. Left to right: gender, age and highest-completed education.

# 6 Attack Efficiency

Having established the baseline performance of the attack that uses the entire dataset, we now focus on its efficiency. In Section 6.1 we study how reducing the amount of data affects the performance of $F_\mathcal{A}$. In Section 6.2 we explore different data augmentation techniques to further reduce the number of queries required.

## 6.1 Variable Amount of Data

For each task, we reduce the amount of data used to extract $F_\mathcal{V}$. We conduct the experiment with randomly sampled $25\%$, $50\%$, $75\%$ of the original attack dataset; each experiment is repeated five times. Figure 7 depicts how the performance of $F_\mathcal{A}$ decreases with less data. We plot both the comparison with $F_\mathcal{V}$'s output (experiment **(B)**) and with the style images (experiment **(C)**).

For Monet-to-Photo, the model does not improve with more than $75\%$. We conjecture that this is enough because the style target style (Monet paintings) is consistent across many images.

For Selfie-to-Anime, the improvement plateaus around $50\%$. Although the mean FID does not improve with more data, the standard deviation is lower at $75\%$. It is the most difficult of the three tasks since there is substantial variability in the anime faces. However, the attack dataset is the largest and it has a lot of redundancy.

For Super-Resolution, the attack improves with more data, up to $100\%$. This suggests that $F_\mathcal{A}$ could be further improved with more data. Although the attack was carried out using a small dataset, the performance is still comparable to $F_\mathcal{V}$

## 6.2 Data Augmentation

In order to improve the efficiency of the attack, $\mathcal{A}$ could craft synthetic queries or choose queries that more effectively explore the decision boundaries. However, such approaches introduce a pattern to the queries, and there exist tech-

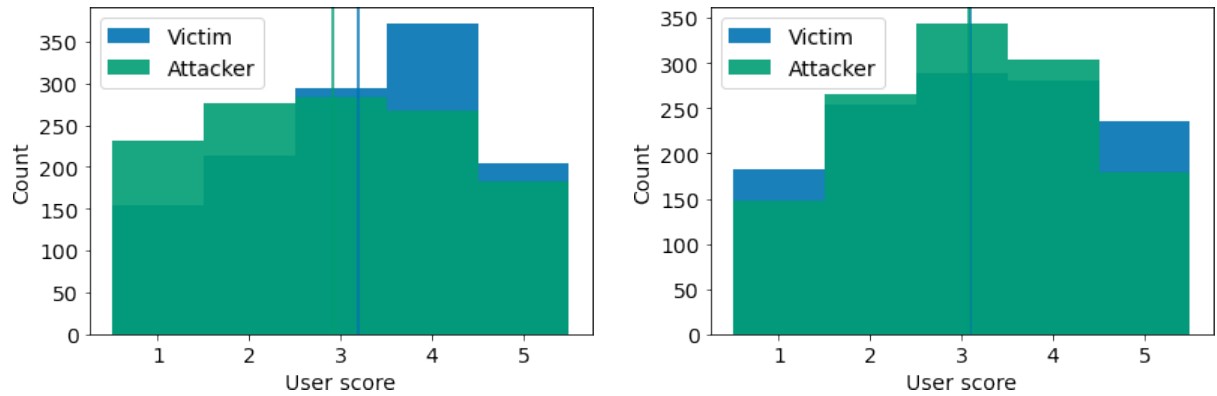

Figure 6: Comparison of scores assigned by the user study participants. **Monet-to-Photo (left)**: for $F_\mathcal{V}$ score mean = 3.20 and std = 1.59, and for $F_\mathcal{A}$ mean = 2.91 and std = 1.76. Vertical lines indicate means of corresponding distributions. **Selfie-to-Anime (right)**: for $F_\mathcal{V}$ score mean = 3.11 and std = 1.76, and for $F_\mathcal{A}$ mean = 3.08 and std = 1.50. Vertical lines indicate means of corresponding distributions.

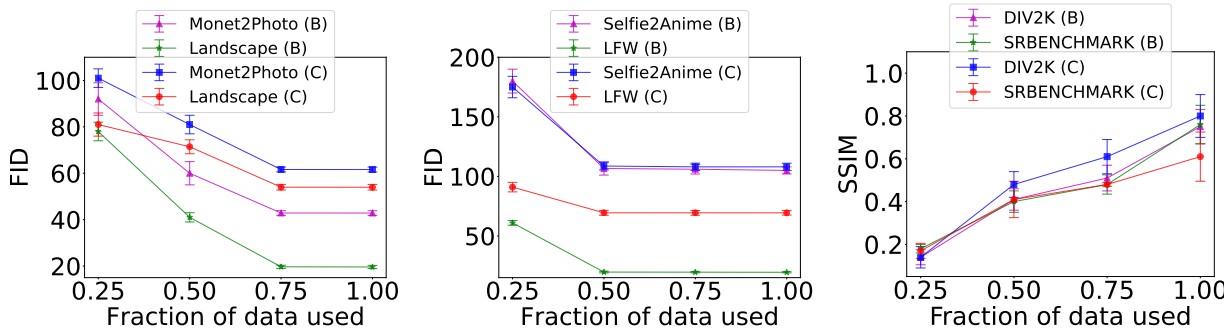

Figure 7: Impact of the amount of data on the effectiveness of the attack. Left to right: Monet-to-Photo, Selfie-to-Anime, Super-Resolution (SSIM). For Monet-to-Photo the performance plateaus at 75%, Selfie-to-Anime stops improving at 50%, Super-Resolution continues to improve up to 100% (we observe the same trend for SSIM and PSNR).

niques that can detect them (Quiring et al., 2018; Juuti et al., 2019; Zheng et al., 2019). Instead, $\mathcal{A}$ can augment their data locally while training $F_\mathcal{A}$. This way, $\mathcal{A}$ improves the query-efficiency of the attack, without risking detection.

In this section, we study the impact of the following augmentation techniques: horizontal flip, rotation (5 degrees), cutout (crop in on a random part of the image), contrast change (20% increase). We do not apply the cutout to Super-Resolution because the images used during training are already small. For each task, we train $F_\mathcal{A}$ with varied amount of data up to the plateau identified in Section 6.1: 25%, 50%, 75% for Monet-to-Photo, 25%, 50% for Selfie-to-Anime, and 25%, 50%, 75%, 100% for Super-Resolution. Each experiment is repeated five times.

In Table 6 we summarise the impact of data augmentation on the performance of $F_\mathcal{A}$. For Monet-to-Photo, horizontal flip (6%) and rotation (4%) improve the performance for $F_\mathcal{A}$s trained with 25% and 50%. These augmentations do not further improve $F_\mathcal{A}$ trained with 75% of data. With cutout, the performance does not improve, and the standard deviation increases. Changing the contrast has no impact on $F_\mathcal{A}$ trained with 75% of the data. However, similarly to the cutout, it increases the standard deviation of $F_\mathcal{A}$s trained with 25% and 50%; one of the five $F_\mathcal{A}$s trained with 25% failed to converge.

For Selfie-to-Anime, horizontal flip (5%) and rotation (2%) improve the performance of $F_\mathcal{A}$s trained with 25% of the data. These augmentations do not further improve $F_\mathcal{A}$ trained with 50%. Cutout hurts the performance of $F_\mathcal{A}$ by about 15% for $F_\mathcal{A}$ trained with 50% of data, and by 20% for $F_\mathcal{A}$ trained with 25%. Similarly to Monet-to-Photo, changing

Table 6: Impact of data augmentation on the performance of $F_\mathcal{A}$s trained with the subset of attack data that does not reach the plateau: Monet-to-Photo 50%, Selfie-to-Anime 25%, and Super-Resolution 75% . For brevity, we present the results for the comparison with the ground truth (experiment **(C)**). We report the mean and standard deviation over five runs.

| Task | Test Set | Baseline (before plateau) | Augmentation | | | |
|------|----------|---------------------------|------|----------|--------|----------|
| | | | Flip | Rotation | Cutout | Contrast |
| Monet-to-Photo | Monet2Photo Test | $81.09 \pm 3.78$ | $76.22 \pm 3.52$ | $77.84 \pm 3.6$ | $81.09 \pm 5.22$ | $80.97 \pm 3.81$ |
| | Landscape | $71.42 \pm 3.01$ | $68.56 \pm 3.01$ | $67.13 \pm 2.89$ | $71.42 \pm 4.67$ | $71.44 \pm 3.11$ |
| Selfie-to-Anime | Selfie2Anime Test | $174.42 \pm 8.89$ | $165.69 \pm 7.68$ | $170.93 \pm 8.41$ | $207.04 \pm 9.13$ | $174.42 \pm 9.24$ |
| | LFW Test | $91.42 \pm 4.04$ | $86.82 \pm 3.97$ | $89.59 \pm 4.00$ | $109.20 \pm 4.07$ | $91.42 \pm 4.92$ |
| Super-Resolution | DIV2K Test | $0.61 \pm 0.08$ | $0.65 \pm 0.07$ | $0.61 \pm 0.07$ | - | $0.61 \pm 0.07$ |
| | SRBENCHMARK | $0.48 \pm 0.01$ | $0.52 \pm 0.01$ | $0.49 \pm 0.01$ | - | $0.48 \pm 0.01$ |

the contrast has no impact $F_\mathcal{A}$ trained with 50% of the data. However, it does increase the standard deviation for $F_\mathcal{A}$ trained with 25%.

For Super-Resolution, horizontal flip improves the performance of $F_\mathcal{A}$ by $4 - 6\%$ for $F_\mathcal{A}$ trained with 25%, 50%, 75% of data; for $F_\mathcal{A}$ trained with 100% the mean performance does not improve but the standard deviation decreases. Rotation does not have any impact for any of the experiments. Changing the contrast, does not affect $F_\mathcal{A}$s trained with 75% and 100% of the data. However, it does increase the standard deviation for $F_\mathcal{A}$ trained with 25% and 50% of the data.

In conclusion, horizontal flip and rotation can improve the performance of $F_\mathcal{A}$. $\mathcal{A}$ can query fewer samples to achieve the same level of performance. However, these augmentations do not improve $F_\mathcal{A}$ beyond the plateau. On the other hand, cutout and changing the contrast at best do not improve the performance, and at worst are detrimental to it. We conjecture that changing the contrast does not maintain the target style anymore and hence, leads to worse performance. Similarly, cutout has minor negative impact on Monet-to-Photo because the style is consistent across the entire image. However, for Selfie-to-Anime contextual information about the face might get lost which leads to a major decrease in performance.

# 7 Evaluation of Potential Defenses

We consider several techniques previously proposed to prevent model extraction in other settings, typically in image classification models, to evaluate their applicability to counter model extraction against image translation models.

## 7.1 Watermarking

**Trigger-based** watermarking is a class of techniques which embed a watermark into the model by modifying a subset of the training samples (e.g. (Zhang et al., 2018; Adi et al., 2018)) or by flipping their corresponding labels (Szyller et al., 2020).

DAWN (Szyller et al., 2020) is a watermarking scheme designed specifically to deter model extraction attacks against classifiers. DAWN changes the prediction label for a small fraction of queries sent by the client and saves them. The query-prediction pairs (trigger set) associated with each client can be then used to verify ownership. $\mathcal{A}$ embeds the watermark while training $F_\mathcal{A}$ with the labels obtained from $F_\mathcal{V}$. To verify ownership, $\mathcal{V}$ queries a suspected $F_\mathcal{A}$ with the trigger set, and declares theft if the predictions match the trigger set.

Although image translation models do not use any labels, we can adapt DAWN (we use the source code provided by the authors[2]) by combining it with approaches that modify the samples (Zhang et al., 2018). For a small fraction of queries (0.5%), we modify the styled image $X_S$, record the pair for future verification, and then return $X_S$ to the client. For a more detailed explanation of the process, see the original paper (Szyller et al., 2020). We experimented with three different kinds of modifications: 1) blurring the image; 2) changing colours to monochrome; 3) adding text (see Figure 8 for examples).

---

[2]https://github.com/ssg-research/dawn-dynamic-adversarial-watermarking-of-neural-networks

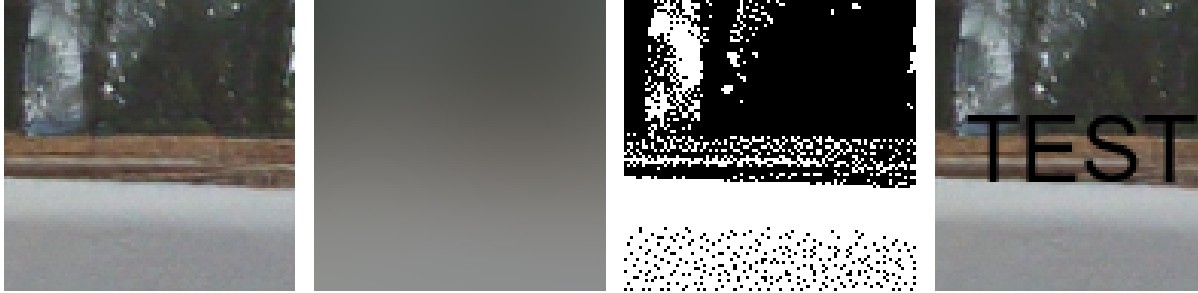

Figure 8: Examples of modifications used to create a trigger set. Left to right: clean sample, Gaussian blur, monochromatic transformation of colours, embedded text.

In all three cases, $F_A$ learns its primary task and the watermark is not embedded. In the initial training stages (5-10 epochs), some watermark artifacts are visible. However as the training progresses, they disappear.

**Steganography-based** fingerprinting is an approach where $V$ can add an imperceptible bit sequence to all training data, instead of embedding a visible watermark. We evaluated one such approach which showed that a GAN trained with fingerprinted data will generate images containing that fingerprint (Yu et al., 2021). To conduct our experiment, we used the source code provided by the authors[3].

$V$ jointly trains an autoencoder that embeds the fingerprint, and a detection network that identifies it. The fingerprint is added to all $X_S$ that are returned to $A$. However, in our experiments the embedding of the fingerprint fails in one of the two ways: 1) the embedding model collapses and produces random pixels, or 2) the fingerprint is not embedded and the detection accuracy is 50% (random guess). We have reached out to the authors but to no avail.

## 7.2 Adversarial Examples and Poisoning

In order to slow down a model extraction attack, $V$ can perturb the output of $F_V$. Typically, this means adding noise to the prediction logit (Lee et al., 2018; Orekondy et al., 2020) in order to *poison* $A$'s training. Similarly to watermarking, image translation models do not use any labels. However, $V$ can inject adversarial noise into $X_S$.

We conducted a baseline experiment to determine if this approach is viable. To each $X_S$, we added a perturbation $\epsilon = 0.25$ (under $\ell_\infty$) using projected gradient descent (Madry et al., 2017). The perturbations were crafted with the discriminator of $F_V$, and using the CleverHans library[4]. These samples do not transfer to $F_A$'s discriminator, and their model trains correctly.

Furthermore, in our experiments, we discovered that adversarial examples crafted using one generator architecture do not transfer to another one (CycleGAN to Pix2Pix) (Figure 9).

Under realistic assumptions, $V$ can craft adversarial examples only using their own models. Creating an ensemble of GANs, each trained separately for the same style transfer task, in order to improve transferability is too costly.

## 8 Discussion

**Metrics.** In the Super-Resolution setting, we have access to ground truth images in the form of original high resolution photos. Hence, the metrics we used for Super-Resolution (PSNR and SSIM) are sufficient for assessing the comparative effectiveness of $F_V$ and $F_A$. This is why a separate user study for Super-Resolution was unnecessary.

For the style transfer settings (Monet-to-Photo and Selfie-to-Anime) there is no such ground truth – given an input image, there is no quantitative means of assessing if the corresponding output image looks like a Monet paining or an anime figure. Our experience with pixel-based metrics (Section 2.3) underscored the possibility that perceptual similarity metrics may not be adequate. This is our motivation for a separate user study to assess the effectiveness

---

[3]https://github.com/ningyu1991/ArtificialGANFingerprints
[4]http://www.cleverhans.io

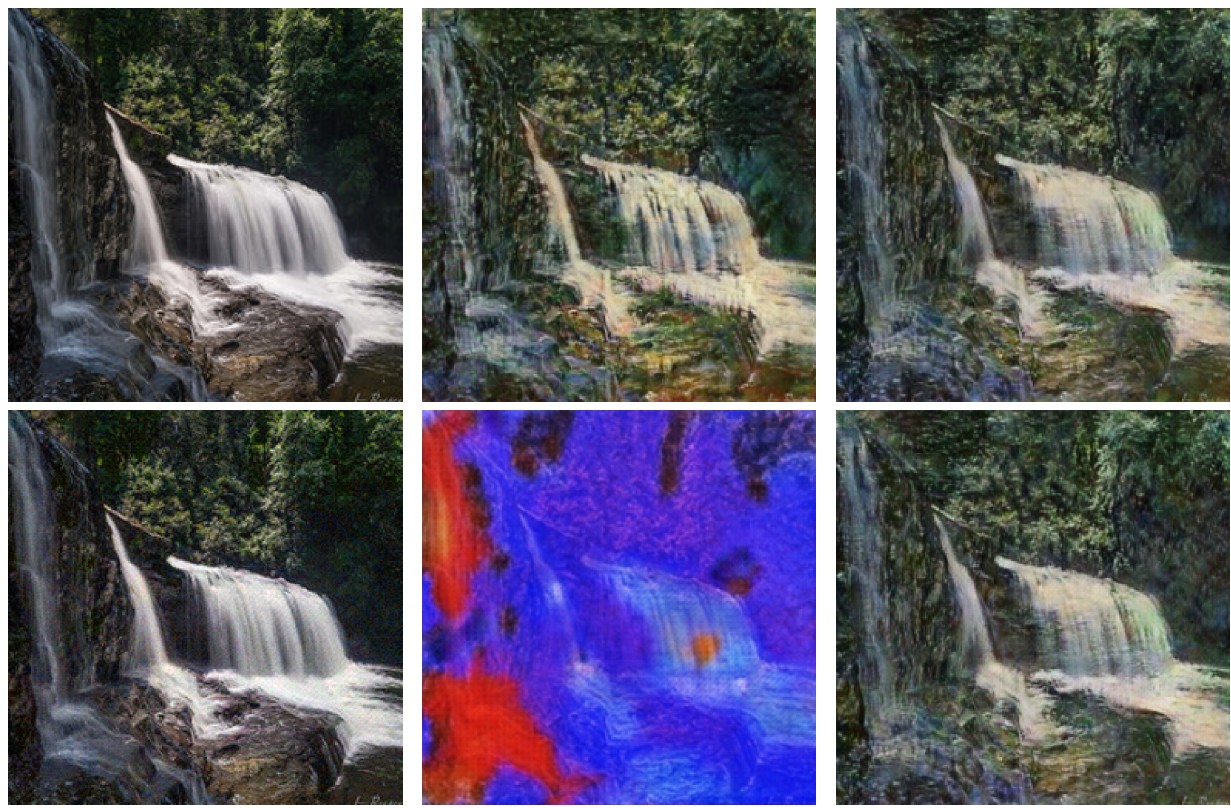

Figure 9: Example translation of an adversarial example crafted using $F_\mathcal{V}$. **Left**: unstyled, **center**: $F_\mathcal{V}$ output (Cy-cleGAN), **right**: $F_\mathcal{A}$ output (Pix2Pix). **Top row**: using a clean sample as input, **bottom row**: using an adversarial example as input.

in the Monet-to-Photo and Selfie-to-Anime settings. The distance metric we used (FID) indicated that the output distributions of $F_\mathcal{V}$ and $F_\mathcal{A}$ are close, as did manual inspection of a random subset of the transformations. The study confirmed that both models are similar within reasonable equivalence bounds. However, the user study indicated that $F_\mathcal{V}$ and $F_\mathcal{A}$ are more similar in the Selfie-to-Anime setting than in the Monet-to-Photo setting (Figure 6) whereas the FID metric indicated otherwise (Tables 3 and 4). This highlights the importance of more robust quantitative metrics for comparing human perception of image similarity.

Additionally, we emphasize that evaluating model extraction attacks against image translation models is more challenging than that of classifiers and NLP models. In all prior work on the extraction of classifiers, average and per-class accuracy, as well as top-k accuracy, were shown to be sufficiently good for measuring the performance of the surrogate model.

Although surrogate NLP models can be evaluated using e.g. BLEU scores, models with similar scores can produce qualitatively different outputs. Hence, they would benefit from user studies. Nevertheless, existing extraction attacks against NLP models (Krishna et al., 2020; Wallace et al., 2021) did not conduct any user studies.

To the best our knowledge, ours is the first work that uses a user study to evaluate the success of model extraction. We are the first to show that image translation models are also vulnerable, and hence, they need to be protected.

**Cost estimate.** To extract the model, $\mathcal{A}$ needs to query $F_\mathcal{V}$ thousands of times. It begs the question if the cost of the extraction attack is prohibitive. We estimate the cost based on the pricing of of OpenAI's models [5] At the resolution of $256 \times 256$, a single query costs \$0.016. Therefore, it would cost approximately \$1250 to extract Selfie-to-Anime model (80,000 queries), \$160 for Monet-to-Photo (10,000 queries), and \$50 for Super-Resolution (2,700 queries).

---

[5]https://openai.com/api/pricing/ Accessed: 2023-01-05.

Note that this pricing corresponds to a much bigger and more powerful text-to-image model, and hence, our estimate is an upper bound on the cost of the attack.

**Towards a defense.** In this work, we emphasize the feasibility of attacking realistic $F_{\mathcal{V}}$, and show that having obtained $F_{\mathcal{A}}$, $\mathcal{A}$ can launch a competing service. The next step would be to explore ways of protecting against such attacks. None of the existing defenses against model extraction attacks explained in Section 2 apply to image translation models. Most of these defenses (Juuti et al., 2019; Atli et al., 2020; Quiring et al., 2018; Kesarwani et al., 2018; Zheng et al., 2019) rely on examining the distribution of queries from clients to differentiate between queries from legitimate clients and adversaries. In our adversary model, we assumed that $\mathcal{A}$ uses natural images drawn from the same domain as $\mathcal{V}$. As a result, none of those defenses are applicable to our attack. Other defenses (Kariyappa & Qureshi, 2019; Orekondy et al., 2020) rely on perturbing prediction vectors which is only applicable to attacks against classifiers.

Alternatively, $\mathcal{V}$ could try to embed a watermark into the model such that all output images contain a trigger that would transfer to $F_{\mathcal{A}}$. This would allow $\mathcal{V}$ to prove ownership and deter $\mathcal{A}$ that wants to launch a competing service but it does not stop the attack on its own. We adapted four existing watermarking schemes (three designed for image classifiers (Szyller et al., 2020; Zhang et al., 2018), one for GANs (Yu et al., 2021)) to black-box extraction of image translation models (Section 7). We show that these schemes do not successfully embed a watermark in $F_{\mathcal{A}}$. Furthermore, recent work showed that *all* model watermarking schemes are brittle (Lukas et al., 2021), and can be removed.

One plausible way to prevent $\mathcal{A}$ from extracting image translation models is to investigate ways of incorporating adversarial examples and data poisoning into a model as a defense mechanism. $\mathcal{V}$ could add imperceptible noise to the output images, designed to make the training of $F_{\mathcal{A}}$ impossible or at least slow it down such that it is not economically viable. However, it was recently shown that such defensive perturbations work only against existing models and are unlikely to transfer to newer and more resilient model architectures (Radiya-Dixit & Tramèr, 2021). Hence, such an approach provides only a temporary defense. Despite this limitation, we evaluated this approach (Section 7). In our experiments, adversarial examples crafted using $F_{\mathcal{V}}$ do not transfer to $F_{\mathcal{A}}$. To overcome this, $\mathcal{V}$ needs to know the architecture of $F_{\mathcal{A}}$, and craft adversarial examples against it (which $\mathcal{A}$ could evade by changing the architecture); or $\mathcal{V}$ could train several $F_{\mathcal{V}}$s using different architectures, and try to craft examples that transfer. Because training multiple, different GANs is expensive this approach is not viable.

**Ethical considerations.** Since this work describes attacks, we do not publicly release our code as usual but will make it available only to bonafide researchers, to facilitate reproducibility. Also, the user study involved human subjects. However, it was conducted according to our institution's data management guidelines (Section 5.4).

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
