# OpenReview forum: "Good Artists Copy, Great Artists Steal: Model Extraction Attacks Against Image Translation Models"
_TMLR — Rejected by TMLR_

### Review · Reviewer_tHtp · 2022-12-02

**Summary Of Contributions:**

The paper presents an extraction attack against image translation models. It follows a well-known strategy that first queries the victim model for data samples and then use them to retrain a surrogate model. The paper evaluates the attack in three settings in image translation: (1) Selfie-to-Anime and (2) Monet-to-Photo, and (3) Super-Resolution (super resolution). Results show that their attack in effective, furthermore, a user study shows human perception of the images produced by original and surrogate models is similar.

**Audience:**

No

**Claims And Evidence:**

Yes

**Requested Changes:**

For this paper to be published at TMLR, authors need to contribute to/improve upon the existing methodology of ''query and retrain", rather than find a new area to which the methodology can be directly applied.

**Strengths And Weaknesses:**

### Major
Unfortunately, the paper does not make any contribution at the conceptual level. The attack follows a well-known concept which has been extensively studied in various problem domains. The only novelty comes from that their attack targets image translation models, which could be the first attempt in literature of model extraction attacks. But this is not even remotely close to meeting the bar of a publication at any top-tier conference/journal.

 ### Minor
- In caption of Figure 1 and 2, the meaning of V and A is not given until section 2.1.
- Second paragraph in Section 2.1 *but may use known techniques...*, authors should give concise description of those techniques with citations rather than just the citations alone.
- Third paragraph in Section 2.1 *Instead, the defender can perturb the prediction vector before...* Here the word "Instead" does not make sense to me, is there anything negative about the techniques you said previously?? If not the word should be **alternatively**
- Last paragraph in Section 2.1, I don't understand this entire paragraph. If the watermark can also transfer to surrogate model, then how can defenders use it to certify the ownership of the original model?
- Section 3, first paragraph, fifth line *to any source style images S*, the notation is not consistent with what the authors use in Section 4 in which S is a template Xs is image.

---

> ### Author Response · Authors · 2022-12-15
> **Our contributions, and clarifications**
>
> > Unfortunately, the paper does not make any contribution at the conceptual level. The attack follows a well-known concept which has been extensively studied in various problem domains. The only novelty comes from that their attack targets image translation models, which could be the first attempt in literature of model extraction attacks. But this is not even remotely close to meeting the bar of a publication at any top-tier conference/journal.
>
> We want to emphasize that we have multiple contributions in our paper:
> 1) we show that image translation models are vulnerable to model extraction attacks:
> 	- the adversary can follow a simple but realistic strategy to obtain a surrogate model,
> 	- surrogate and victim models are equivalent according to human judgement,
> 	- we show how the amount of data affects the effectiveness of the attack, and how to improve the efficiency of the attack through local data augmentation.
> 2) we study established defenses against model extraction of classifiers, and show that they cannot be used to protect image translation models.
>
> Since image translation models were not previously shown to be vulnerable, we believe that the community of ML researchers and practitioners would benefit from our findings. It is particularly important now due to the publicity that generative models receive in the media.
>
> Given our contributions and the relevance to the community, our paper meets the evaluation criteria put forward by the TMLR committee.
>
> > Last paragraph in Section 2.1, I don't understand this entire paragraph. If the watermark can also transfer to surrogate model, then how can defenders use it to certify the ownership of the original model?
>
> We follow the DAWN watermarking scheme introduced in (Szyller et al. 2020). It is designed specifically against model extraction attacks. The watermark is not embedded in $F_V$ but in $F_A$: $V$ modifies the output for a subset of the queries they receive, and records them (trigger set), $A$ trains $F_A$ using those outputs, the watermark gets embedded in $F_A$. To verify ownership, $V$ queries exposed $F_A$ with the trigger set.
>
> -> We will update the description of the watermarking scheme to make it clearer.
>
> Minor:
>
> -> We will address the remaining minor comments.

---

### Review · Reviewer_BtNP · 2022-12-27

**Summary Of Contributions:**

In this work, the authors conduct model extraction attacks on style transfer GANs when the adversary has access to the model's API. They focus on three different types of style transfer models: Selfie-to-Anime, Monet-to-Photo (image style transfer), and Super-Resolution. To evaluate the effectiveness of their approach, they use both metric-based evaluation and a user study. This work provides insights into the vulnerabilities of style transfer GANs to model extraction attacks.


**Audience:**

Yes

**Broader Impact Concerns:**

No ethical concerns

**Claims And Evidence:**

No

**Requested Changes:**

* Cost analysis based on the number of api call
* Experiments on a larger dataset
* more reasoning why the current user study means model extractions

**Strengths And Weaknesses:**



One notable aspect of this work is that the authors have chosen to apply their approach to a real-world application, demonstrating the relevance and importance of their findings in practical settings. This choice adds credibility to their work and helps to underscore the potential implications of their results for real-world systems.

Model extraction attacks that utilize API access often involve making a large number of calls to the API in order to gather enough data to reconstruct the model. While this work does evaluate the impact of the size of the training dataset on the effectiveness of model extraction, it does not examine the effect of the number of calls made to the API. This is an important factor to consider, as it may significantly affect the feasibility and cost of such attacks. It would be beneficial for the authors to consider this aspect in their evaluation and to consider how the number of calls may impact the practicality of their approach.

One potential concern with this work is how well it would scale to situations where the size of the training dataset is larger. The tasks that the authors have focused on in this work involve relatively small datasets, and they make use of a large number of queries in their model extraction attacks. It is unclear how well these results would generalize to cases where the training dataset is much larger, and it would be useful for the authors to consider this question and to consider how their approach might perform in these scenarios.


The defenses considered in this work may not be comprehensive enough to fully address the threat of model extraction attacks. For example, the authors could consider using traditional steganography techniques or simple copyright watermarking as defenses. While these approaches may not be foolproof, they could potentially provide some level of copyright  protection against model extraction attacks and would be worth exploring as potential defense strategies. It would be useful for the authors to consider a wider range of defense techniques in order to more fully understand the potential vulnerabilities and countermeasures in this domain.

Based on the information provided, it seems that the results of the model extraction attacks are mainly demonstrating the transfer of knowledge rather than the successful extraction of the model itself. This is reflected in the relatively high Fréchet Inception Distance (FID) scores observed when comparing the V distribution to other distributions or benchmarks.

The user survey conducted in this work asks users to determine whether the generated images are from the target transfer style or not. While this is an important aspect of the evaluation, it may not be sufficient to fully assess the success of the model extraction attacks. To truly gauge the effectiveness of these attacks, it would be necessary for users not to be able to distinguish between images generated by the target model and those generated by the reconstructed model. This would provide a more accurate assessment of the degree to which the model has been successfully extracted and the potential consequences of such an attack.

---

> ### Author Response · Authors · 2023-01-05
> **Response and action points (part I)**
>
> > While this work does evaluate the impact of the size of the training dataset on the effectiveness of model extraction, it does not examine the effect of the number of calls made to the API. This is an important factor to consider, as it may significantly affect the feasibility and cost of such attacks. It would be beneficial for the authors to consider this aspect in their evaluation and to consider how the number of calls may impact the practicality of their approach.
>
> In our experiment setup, the size of the dataset used by the adversary is the number of calls (queries) to the API. If you’re referring to some monetary cost associated with querying the API, we could give an estimate for different price brackets.
>
> For example, DALLE2 is priced at 0.016 USD/image (at the resolution of 256x256). It would cost approximately 1250 USD to extract the selfie2anime model (80k queries), and 160 USD for the monet2photo (10k queries). Note that this is an upper bound since DALLE is a much bigger and more powerful type of model that the image translation models in our work.
>
> -> We will add a cost estimate for the attacks.
>
> > One potential concern with this work is how well it would scale to situations where the size of the training dataset is larger. The tasks that the authors have focused on in this work involve relatively small datasets, and they make use of a large number of queries in their model extraction attacks. It is unclear how well these results would generalize to cases where the training dataset is much larger, and it would be useful for the authors to consider this question and to consider how their approach might perform in these scenarios.
>
> Note that in our work, the attacker is interested in extracting the model s.t. they can **duplicate the style $S$ in the surrogate model**. Hence, the difficulty of running the attack is more related to the style, rather than the size of the style dataset. Additionally, note that in image translation tasks, many of the style datasets are small in size due to the limited number of source material, e.g. paintings of a given author, or characters of a particular cartoon show.
>
> > The defenses considered in this work may not be comprehensive enough to fully address the threat of model extraction attacks. For example, the authors could consider using traditional steganography techniques or simple copyright watermarking as defecses. While these approaches may not be foolproof, they could potentially provide some level of copyright protection against model extraction attacks and would be worth exploring as potential defense strategies. It would be useful for the authors to consider a wider range of defense techniques in order to more fully understand the potential vulnerabilities and countermeasures in this domain.
>
> We investigated defences that were shown (Szyller et al. 2020, Lee et al. 2018, Orekondy et al. 2020) to be effective against some model extraction attacks, and showed that they cannot be easily adapted to the image translation domain.
>
> Although we can extend the discussion, including the topics that you mentioned, we cannot recommend any of them as a viable defence to the ML community. Steganographic techniques are not resilient to post-processing which is part of the regularisation and training during the attack. Additionally, for the surrogate model to learn the embedded code, it would need to be the same across all $X_S$, which would make it detectable by the adversary.
>
> We emphasise that currently there is no obvious way to defend against the extraction of image translation models. Designing a new defence requires extended analysis and experimentation, and we leave it out as future work.

---

> ### Author Response · Authors · 2023-01-05
> **Response and action points (part II)**
>
> > Based on the information provided, it seems that the results of the model extraction attacks are mainly demonstrating the transfer of knowledge rather than the successful extraction of the model itself. This is reflected in the relatively high Fréchet Inception Distance (FID) scores observed when comparing the V distribution to other distributions or benchmarks.
>
> and
>
> > The user survey conducted in this work asks users to determine whether the generated images are from the target transfer style or not. While this is an important aspect of the evaluation, it may not be sufficient to fully assess the success of the model extraction attacks. To truly gauge the effectiveness of these attacks, it would be necessary for users not to be able to distinguish between images generated by the target model and those generated by the reconstructed model. This would provide a more accurate assessment of the degree to which the model has been successfully extracted and the potential consequences of such an attack.
>
> In our work, we focus on __functionally equivalent__ extraction attacks rather than __exact extraction__ that you are describing. These two classes of attacks typically have different requirements and goals for the attack. The goal of functionally equivalent extraction is to obtain models that offer the same functionality to the user but do not need to be exactly the same.
>
> In other words, yes, you are correct in saying that during our attack is to transfer-learn the secret style. As a result, the goal of the study was to verify that the participants think that the models offer the same **quality of transformation**, even if the transformations done by $F_V$ and $F_A$ are different.

---

### Review · Reviewer_bynz · 2022-12-30

**Summary Of Contributions:**

This paper proposes a framework for stealing GANs for image translation tasks. The paper mostly focuses on two applications: style transfer and super resolution. It assumes that we can query the victim models as many times as needed to create training datasets for training surrogate ones. The authors perform quantitate and qualitative evaluations to evaluate the proposed approach and provide promising results. Traditional defenses like watermarking and data poisoning are shown not effective against the approach.

**Audience:**

Yes

**Broader Impact Concerns:**

No broader impact concerns

**Claims And Evidence:**

Yes

**Requested Changes:**

- Data efficiency is important from my understanding. The paper assumes unlimited query access to the victim model but more query means higher chance to be detected. It would be a plus to have more novel design/discuss in using less data for training surrogate models besides data augmentation. For instance, X_A can be selected according to each step of the trained surrogate models. Also, one can pretrain with public datasets and only finetune with the query data from victim models.
- The proposed method in Sec 4 can be framed into a more clear, formal, and general way.
- There exists many works on stealing image classifiers and NLP models. Why stealing image translation GANs is hard compared to other tasks should be emphasized to better understand the contributions of the proposed approaches.

**Strengths And Weaknesses:**

### Strength
- The proposed approach is new and interesting. Studies on how to steal image translation models and new defend approaches against them are urgently needed for industry.
- The paper performs extensive evaluation with promising results provided.
- Empirical attack efficiency study is useful.
- The paper shows that existing defenses like watermarking do not work for the proposed approach.

Weakness
- The proposed approach is simple and has limited novelty.
- Presentation on the insights of the proposed approach can be improved (see changes below).

---

> ### Author Response · Authors · 2023-01-05
> **Response and action points**
>
> > Data efficiency is important from my understanding. The paper assumes unlimited query access to the victim model but more query means higher chance to be detected. It would be a plus to have more novel design/discuss in using less data for training surrogate models besides data augmentation. For instance, X_A can be selected according to each step of the trained surrogate models. Also, one can pretrain with public datasets and only finetune with the query data from victim models.
>
> The adversary can distribute the queries among different clients and run a Sybil attack. Hence, limiting the number of queries for each client does not prevent the extraction. Our attack uses data from the same domain as the victim model, and is indistinguishable from queries made by the benign clients.
>
> Any kind of clever querying introduces a querying pattern into the attack that could be detected. Model extraction attacks against classifiers that rely on synthetic data, or selected queries were shown to be detectable. As discussed in Section 6, we believe that local augmentation improves the efficiency without increasing the detectability of the attack (Quriring et al. 2018, Juuti et al. 2019, Zheng et al. 2019).
>
> Lastly, if there exist suitable public datasets to pretrain a given image translation model, then the adversary can use them. However, in our work we argue that the style template that the adversary would want to copy is secret; therefore, there would not be any public data.
>
> -> We will extend the discussion in Section 6.
>
> > The proposed method in Sec 4 can be framed into a more clear, formal, and general way.
>
> Our goal was to present the general methodology for extracting image translation models.
>
> In our proposed method, the adversary makes no assumption about how the victim model is trained or its architecture. The adversary relies exclusively on the query access, and knowing the victim domain. Therefore, our technique is generally applicable to any image translation model.
>
> -> We will update Section 4 to clarify its generality.
>
> Are there any other clarifications that you would like to see?
>
> > There exists many works on stealing image classifiers and NLP models. Why stealing image translation GANs is hard compared to other tasks should be emphasized to better understand the contributions of the proposed approaches.
>
> Although prior work showed that other types of models (such as image classifiers) are susceptible to model extraction attacks, evaluating whether a model extraction technique is successful is straightforward in those settings (for example, a successful extraction of an image classifier should result in a surrogate model whose accuracy is comparable to that of the victim model). In all prior work on extraction of classifiers, average and per-class accuracy, as well as top-k accuracy, were shown to be sufficiently good for measuring the performance of the surrogate model.
>
> However, In our work, we emphasise that the evaluation of surrogate translation models is challenging; commonly used metrics (e.g. FID) do not capture the similarity of transformations that are both **qualitatively good**, albeit different. Hence, extraction of image translation models requires subjective human inspection: this was our motivation for conducting the user study.
>
> Models in NLP, such as large language models that generate text, also share the same challenge. Evaluation of surrogate NLP models can rely on standard metrics (e.g. BLEU scores). However, NLP models with similar BLEU scores can give qualitatively different outputs, and hence, would benefit from user studies. Nevertheless, existing extraction attacks against NLP models (Krishna et al. 2020, Wallace et al. 2021) did not conduct any user study.
>
> To the best our knowledge, ours is the first work that uses a user study to evaluate the success of model extraction. We are the first to show that image translation models are also vulnerable, and hence, they need to be protected.
>
> -> We will extend the discussion to emphasize this aspect.

---

### Author Response · Authors · 2023-01-09
**Summary of the changes**

Thank you for your constructive feedback.

We have updated the paper based on the discussion so far. Below, we list each change we proposed (prefixed with a “->“) to address reviewers' concerns, and explain how the change was made in the paper.

**Reviewer bynz**

-> We will extend the discussion in Section 6.

We extended the discussion to clarify the risk associated with detectable querying patterns.

-> We will update Section 4 to clarify its generality.

We added a paragraph explaining how our attack applies to other image translation settings.

-> We will extend the discussion to emphasize this aspect.

We extended the discussion point about the metrics (Section 8), to highlight the difficulty of extracting image translation models.

**Reviewer BtNP**

-> We will add a cost estimate for the attacks.

We added a discussion about the estimated cost of the attack in Section 8.

**Reviewer tHhp**

-> We will update the description to make it clearer.

We updated the description in Section 2, and Section 7.

-> We will address the remaining minor comments.

We changed the phrasing and updated some descriptions following your suggestions. Additionally, we added a notation table, and a new symbol $S_X$, to differentiate between: a style template $S$ (e.g. Monet’s style), source images that contain it $S_X$ (e.g. Monet’s paintings), and images styled by the model to apply that style $X_S$ (or $X_{A_S}$).

---

### Decision · Action_Editors · 2023-04-13

**Recommendation:** Reject

**Comment:**

The authors develop extraction attacks against image translation GANs based solely on API access. While reviewers were concerned about the data efficiency of model extraction, I think that demonstrating a successful model extraction attack is sufficiently interesting to at least some members of the TMLR audience.

My main concern about the paper is regarding the user study (as mentioned in the comments on claims and evidence). In particular, I fine the following problematic:
1. "We explained that the study aims to evaluate which of the shown style transformations are successful in the participant’s opinion, and does not evaluate their skill in making this judgement." - This statement is made without any justification. I do not think it is obvious how to design a study that captures the participant's opinion on whether specific nuanced style transformations are successfully made, versus whether the participants themselves are capable of telling this for themselves. Further, the authors do not provide any information on the confidence of judgements made by the participants beyond the Likert scale - since this can be a rather subjective task, I feel a proper user study should include a discussion on confidence of participants' response.
2. The use of statistical tests: In the t-test, the authors pose the problem of judging whether the extracted model is functionally equivalent to the true model by setting up a statistical test to reject the null hypothesis of models being equivalent. However, this is not standard practice: The onus is on the authors to demonstrate equivalence, so the null hypothesis should reflect the contrasting outcome (non-equivalence) so that rejection of the null hypothesis supports the authors' claim. This is indeed the case for the two-sided test, but it is confusing why the authors propose two tests, and further, do not provide any details around correcting for multiple hypothesis testing on the same population.

If the authors can address the above issues and submit a significant revision with a clearly motivated user study, sound statistical analysis of the results and a documentation of the limitations of the user study and unaddressed issues in evaluating functionally equivalent model extraction, I would be happy to reconsider the paper for publication.

**Audience:**

The paper develops successful model extraction attacks against GANs given API access. While the method is not shown to be scalable to larger datasets, it would be of interest to certain members of the TMLR audience to learn about this possibility and build on the work of the authors.

**Claims And Evidence:**

The authors develop a technique for model extraction given API access to image translation GANs. The authors demonstrate that given unlimited query access, they can effectively extract image translation models, showing that the API access is sufficient for model extraction, as measured by automated metrics as well as by a user study.

While the reviewers appreciated the work done by the authors for the user study, the scope of the study (in terms of number of participants, the way the participants were recruited, and the results from the statistical significance analysis) is not sufficient to make a strong claim about the effectiveness of the method proposed.